# Faricimab for Diabetic Macular Edema in Patients Refractory to Ranibizumab or Aflibercept

**DOI:** 10.3390/medicina59061125

**Published:** 2023-06-11

**Authors:** Hiromi Ohara, Yosuke Harada, Tomona Hiyama, Ayako Sadahide, Akira Minamoto, Yoshiaki Kiuchi

**Affiliations:** Department of Ophthalmology and Visual Science, Graduate School of Biomedical Sciences, Hiroshima University, Hiroshima 734-8551, Japanykiuchi@hiroshima-u.ac.jp (Y.K.)

**Keywords:** diabetic macular edema, faricimab, ranibizumab, aflibercept, vascular endothelial growth factor, angiopoietin-2

## Abstract

*Background and Objectives*: Faricimab is the first intravitreal injection of vascular endothelial growth factor-A and angiopoietin-2 bispecific monoclonal antibody. Here, we evaluate the functional and anatomical outcomes of faricimab treatment in patients with diabetic macular edema (DME) that was refractory to ranibizumab or aflibercept. *Materials and Methods*: We performed a retrospective, observational, consecutive-case study of patients who had DME that was refractory to treatment with ranibizumab or aflibercept and were treated with faricimab between July 2022 and January 2023 under a pro re nata regimen. All the participants were followed for ≥4 months after the initiation of faricimab. The primary outcome was a recurrence interval of ≥12 weeks, and the secondary outcomes were the changes in best-corrected visual acuity (BCVA) and central macular thickness (CMT). *Results*: We analyzed 18 eyes of 18 patients. The mean recurrence interval of previous anti-VEGF injection was 5.8 ± 2.5 weeks, which was significantly extended to 10.8 ± 4.9 weeks (*p* = 0.0005) by the switch to faricimab. Eight patients (44.4%) achieved a recurrence interval of ≥12 weeks. A history of subtenon injection of triamcinolone acetonide (*p* = 0.0034) and the presence of disorganization of the retinal inner layers (*p* = 0.0326) were found to be significantly associated with a recurrence interval of <12 weeks. The mean BCVAs were 0.23 ± 0.28 logMAR and 0.19 ± 0.23 logMAR, and the mean CMTs were 473.8 ± 222.0 µm and 381.3 ± 219.4 µm at baseline and 4 months, respectively, but these changes were not statistically significant. None of the patients experienced serious adverse events. *Conclusions*: Faricimab may extend the treatment interval for patients with DME that is refractory to ranibizumab or aflibercept. DME previously treated with the subtenon injection of triamcinolone acetonide or associated with disorganization of the retinal inner layers may be less likely to be associated with a longer recurrence interval after switching to faricimab.

## 1. Introduction

Diabetic macular edema (DME) causes vision impairment in patients with diabetes. The number of adults worldwide with clinically significant DME was estimated to be 18.8 million in 2020 and is expected to increase to 28.6 million by 2045 [1]. The intravitreal of anti-vascular endothelial growth factor (VEGF) injections has been shown to significantly improve functional and anatomical outcomes in several randomized controlled trials, and it has been used as a standard treatment for center-involved DME [2,3]. However, some patients respond poorly to anti-VEGF therapy, and approximately 30% of DME patients had persistent DME after 1 year of sufficient treatment [2,4]. In addition, the number of injections that is typically administered in clinical practice is lower than in clinical trials, and this is associated with a suboptimal visual outcome [5,6].

Faricimab is the first intravitreal injection of VEGF-A and angiopoietin-2 (Ang-2) bispecific monoclonal antibody. It has been shown to be non-inferior to ranibizumab or aflibercept for the improvement of visual acuity and central macular thickness (CMT). Theoretically, faricimab may have a superior effect on DME that insufficiently responds to ranibizumab or aflibercept, owing to its ability to inhibit Ang-2 [7,8]. Therefore, the purpose of this study was to evaluate the efficacy of faricimab in patients with refractory DME.

## 2. Materials and Methods

### 2.1. Study Population

We performed a retrospective study of patients with DME who were treated with faricimab at Hiroshima University Hospital between July 2022 and January 2023 and had previously received anti-VEGF therapy (ranibizumab or aflibercept) but were refractory to these therapies. All the participants were ≥20 years old and were followed for ≥4 months after their first faricimab injection. Four retina specialists (H.O., Y.H., H.T., and A.S.) made the decision to switch patients with DME who demonstrated inadequate resolution or an increase in exudative fluid during ranibizumab or aflibercept therapy or who experienced recurrence within 8 weeks to faricimab. Anti-VEGF or faricimab injections were administrated pro re nata. We measured the CMT and evaluated the structural change in the macular during each visit to determine whether further faricimab injections were required. The criteria for administering additional injections were as follows: (1) an increase in CMT of ≥10% over the minimum value achieved following the switch to faricimab or (2) the presence of intraretinal or subretinal fluid. The recurrence interval was defined as the period between the previous faricimab injection and the time point at which the criteria for the administration of an additional injection were met. The minimum recurrence period during the present study was 4 weeks, which is consistent with the minimum injection interval for patients with DME. The follow-up intervals were determined by each physician on the basis of this recurrence interval and had a range of 4–10 weeks. The visit intervals were reduced or extended by 2–4 weeks on the basis of the recurrence interval. No change or increase in CMT after an injection was defined as a 4-week recurrence period. To minimize bias and to avoid missing subtle retinal changes, two retinal specialists (H.O. and Y.H.) independently evaluated CMT and retinal structural changes using optical coherence tomography (OCT). Patients with any other disease involving the macula (i.e., age-related macular degeneration, retinal vein occlusion, or ocular trauma) were excluded from the study. If both eyes met the inclusion criteria, only the eye that was first treated with faricimab was included. The study complied with the tenets of Declaration of Helsinki and was approved by the Institutional Review Board of Hiroshima University. The requirement for informed consent was waived by the review board, owing to the retrospective nature of the study.

### 2.2. Objectives and Data Collection

The primary outcome was a recurrence interval of ≥12 weeks, and the secondary outcomes were the changes in best-corrected visual acuity (BCVA) and CMT, specifically, an improvement of ≥0.1 logMAR in BCVA and a reduction of ≥20% in CMT between baseline and 4 months. Any serious adverse event was recorded.

The following data were collected: the age at baseline when faricimab was first administered; sex; BCVA, CMT, and intraocular pressure at baseline after 1 and 4 months and at the final visit; the type of diabetes; the duration of known diabetes; the hemoglobin A1c level; a history of hypertension, hyperlipidemia, or chronic kidney disease (estimated glomerular filtration rate (eGFR) < 60 mL/min/1.73 m^2^); lens status (phakia or pseudophakia); the number, recurrence interval, and type of anti-VEGF injections administered prior to faricimab; any other treatments administered prior to switching (focal macular laser, subtenon injection of triamcinolone acetonide (STTA), and pars plana vitrectomy); the presence of vitreous traction and/or epiretinal membranes; any change in the disorganization of the retinal inner layers (DRIL) and/or the ellipsoid zone in a 1 mm diameter foveal area at baseline; the number of and recurrent interval of faricimab injections; and any serious adverse events. Decimal BCVA was assessed using the logarithm of the minimum angle of resolution (logMAR). Spectral-domain OCT was performed using an RTVue XR Avanti (Optovue Inc., Fremont, CA, USA). We defined CMT as the thickness between the internal limiting membrane and the retinal pigment epithelium.

### 2.3. Statistical Analyses

The characteristics of the participants and the details of their treatment were compared between groups according to the recurrence interval using the χ^2^ test, Fisher’s exact test, or Wilcoxon–Mann–Whitney test. Dunnett’s test was used to compare BCVA, CMT among time points (baseline, 1 and 4 months after the first faricimab injection, and the final visit), and treatment interval. JMP software, version 16 (SAS Inc., Cary, NC, USA), was used for the statistical analyses, and *p* < 0.05 was considered to represent statistical significance.

## 3. Results

Twenty-one patients (twenty-five eyes) were treated with faricimab during the study period. Four patients received the treatment in both eyes, and three eyes with a follow-up period of <4 months were excluded from the analyses. Therefore, a total of 18 eyes with DME were included in the present study. Table 1 shows the characteristics of the participants and the previous treatments of DME. All the participants had type 2 diabetes. The mean recurrence interval of previous anti-VEGF injections was 5.8 ± 2.5 weeks (range, 4–13 weeks), and the mean duration of follow-up was 6.1 ± 1.3 months (range, 4–8.5 months).

The mean number of faricimab injections was 2.8 ± 1.4 times (range, 1–6 times). The mean recurrence interval was 10.8 ± 4.9 weeks (range, 4–20 weeks), which was significantly prolonged by the switch to faricimab (*p* = 0.0005). Fourteen patients (77.8%) had a longer recurrence interval compared to previous anti-VEGF therapy, and eight (44.4%) achieved a recurrence interval of ≥12 weeks. However, three participants (16.7%) retained a recurrence interval of 4 weeks. A history of STTA (*p* = 0.0034) and the presence of DRIL (*p* = 0.0326) were found to be significantly associated with a recurrence interval of <12 weeks (Table 2). The mean BCVAs were 0.23 ± 0.28 logMAR, 0.22 ± 0.28 logMAR, 0.19 ± 0.23 logMAR, and 0.19 ± 0.22 logMAR at baseline, after 1 and 4 months, and at the final visit, respectively, after switching to faricimab, but the subsequent BCVA values did not significantly differ from baseline (*p* = 0.9990, *p* = 0.9396, and *p* = 0.9270, respectively). Four participants (22.2%) at the 4-month time point and six (33.3%) at the final visit showed improvements in BCVA of >0.1 logMAR.

The mean CMTs were 473.8 ± 222.0 µµm, 382.2 ± 215.3 µµm, 381.3 ± 219.4 µm, and 327.6 ± 153.0 µm at baseline, after 1 and 4 months, and at the final visit, respectively, which were not statistically significant from baseline (*p* = 0.3999, *p* = 0.3922, and *p* = 0.0903, respectively). Ten participants (55.6%) at 4 months and twelve (66.7%) at the final visit showed reductions in CMT of >20% from baseline.

None of the participants experienced a serious adverse event, including endophthalmitis, intraocular inflammation, or an elevation of intraocular pressure.

## 4. Discussion

In this retrospective study, the mean recurrence interval (defined as an increase in CMT or the appearance of new exudative changes since the preceding visit) was significantly prolonged after switching to faricimab from previous anti-VEGF injections, and more than 40% of the eyes achieved a recurrence interval of ≥12 weeks. The mean BCVA and CMT showed non-significant improvements 4 months after the baseline. In addition, 22.2% of the patients showed improvements in BCVA of ≥0.1 logMAR and 55.6% patients showed improvements in CMT of ≥20% between baseline and 4 months.

According to previous reports, approximately 30% of patients have DME that is resistant to sufficient treatment for 1 year [2,4]. Even after 3 years of monthly or bi-monthly aflibercept treatment, 13% of DME patients continued to require frequent dosing [9].

Faricimab is the first intravitreal injection of a bispecific monoclonal antibody (targeting both VEGF-A and Ang-2) for the treatment of eyes with DME or age-related macular degeneration. The angiopoietin/tyrosine kinase with immunoglobulin and epidermal growth factor homology domains signaling pathways regulate vascular stability, angiogenesis, and inflammation; by inhibiting both molecules, faricimab may have superior effects to other treatments that target only VEGF [10,11]. In the BOULEVARD trial, faricimab was found to significantly improve mean BCVA after 24 weeks in patients with treatment-naïve DME, and they required less frequent re-treatment than patients being treated with ranibizumab [7]. In the YOSEMITE and RHINE studies, faricimab was found to be non-inferior to aflibercept in the improvements in visual acuity and mean CMT after 1 year, with more than 70% of the patients treated with faricimab achieving a treatment interval of ≥12 weeks [8]. In the present study, 44.4% of the participants achieved a recurrence interval of ≥12 weeks, despite the inclusion of patients with refractory DME. Rush et al. reported that patients who switched from aflibercept to faricimab showed significant improvements in both visual acuity and CMT compared with those who continued treatment with aflibercept for persistent DME [12]. In their study, 37.5% of the patients achieved a CMT < 300 µm, and 41.7% of patients showed an improvement of >0.2 logMAR 4 months after switching to faricimab. These findings suggest that faricimab may improve the visual and anatomical outcomes of patients with fewer injections than are necessary using the previous anti-VEGF injections. This may reduce the financial burden, the necessity for multiple visits and injections, the probability of infection, and the systemic complications associated with the treatment.

Furthermore, we found that previous STTA treatment was a risk factor for a recurrence interval of <12 weeks. Of the eyes studied, 61.1% had received STTA, which is similar to the proportion of patients who received STTA in a previous study (64.9%) [6]. Diabetic retinopathy is a multifactorial disease that is associated with not only VEGF, but also other inflammatory cytokines, such as monocyte chemoattractant protein-1, intercellular adhesion molecule 1, interleukin-6, and platelet-derived growth factor [13,14,15]. These cytokines cause chronic inflammation, which leads to leukocytosis, greater vascular permeability, and dysfunction of the blood–retina barrier [16]. In the DRCR.net Protocol U study, the group administered both dexamethasone and ranibizumab was shown to benefit from significantly larger functional and anatomical improvements than the group administered ranibizumab alone [17]. The present results also suggest that DME in the eyes that previously received STTA may have been largely caused by inflammatory cytokines, rather than vascular permeability caused by the VEGF or Ang-2 signaling pathways. The DRIL was also associated with a shorter recurrence interval in the present study. There have been several studies which have shown that the presence and greater severity of DRIL were associated with a poor BCVA response to anti-VEGF injections [18,19]. DRIL could suggest a destruction of bipolar, amacrine, or horizontal cells [20], which may relate to vascular instability or hypervascularity and possibly result in poor response to anti-VEGF injections.

There were several limitations to the present study. First, it was a single-center, retrospective, observational study of a small number of eligible patients. In addition, the duration of follow-up was relatively short, which may have affected our ability to evaluate the long-term effectiveness of faricimab. It is important to note that faricimab was released in Japan on 25 May 2022 and was prescribed at Hiroshima University Hospital starting on 10 July 2022. Consequently, the number of eligible patients was small because of the short timeframe during which faricimab was available for administration. Second, because of the retrospective nature of the study, we were unable to establish a control group of patients who underwent continued treatment with ranibizumab or aflibercept for a direct comparison of treatment outcomes with eyes that were switched to faricimab. Third, the treatment decisions and the visit or treatment interval depended on the preferences of each of the ophthalmologists and patients, which may have introduced selection bias regarding the use and frequency of anti-VEGF or faricimab injections. Finally, the duration of DME, treatment history, number of anti-VEGF injections, and visit/injection intervals varied among the participants, making it impossible to directly compare the efficacy of faricimab with anti-VEGF therapies. Therefore, we could only compare the time to relapse following injection as the primary endpoint to evaluate the efficacy of faricimab.

## 5. Conclusions

Faricimab administration may be associated with a prolongation of the treatment interval in patients with refractory DME. However, patients previously treated with STTA and those with DRIL may be less likely to have longer recurrence intervals after switching to faricimab. Longer-term administration may improve the vascular stabilization effects of Ang-2 and is expected to improve the functional and anatomical status. Further long-term studies are required to confirm this.

## Figures and Tables

**Table 1 medicina-59-01125-t001:** Clinical characteristics of patients at baseline.

	Total (*n* = 18)
Age (years) (mean ± SD)	68.8 ± 9.8
Sex, male/female	7/11
Duration of diabetes (years) (mean ± SD)	18.47 ± 13.2
HbA1c (%) (mean ± SD)	7.6 ± 1.1
Concomitant hypertension, +/–	9/9
Concomitant hyperlipidemia, +/–/unknown	5/5/8
Concomitant of chronic kidney disease, +/–/unknown	4/5/9
Lens status, phakia/pseudophakia	4/14
BCVA (mean ± SD)	0.23 ± 0.28
IOP (mean ± SD)	16.2 ± 2.9
CMT (mean ± SD)	473.8 ± 222.0
DR status, mild NPDR/moderate NPDR/severe NPDR/PDR	4/1/9/4
Duration of DME (months) (mean ± SD)	28.1 ± 23.4
Number of previous anti-VEGF injections (mean ± SD)	5.7 ± 5.2
Type of previous anti-VEGF injections, ranibizumab/aflibercept/both	3/9/6
Complications with vitreous traction and/or epiretinal membranes	9 (50.0%)
DRIL	7 (38.9%)
Disorganization of ellipsoid zone	10 (55.6%)
Prior treatments other than anti-VEGF therapy	16 (88.9%)
Subtenon injections of corticosteroids	11 (61.1%)
Focal macular laser	10 (55.6%)
Pars plana vitrectomy	3 (16.7%)

DME: diabetic macular edema, SD: standard deviation, HbA1c: glycated hemoglobin, BCVA: best-corrected visual acuity, logMAR: logarithm of the minimum angle of resolution, IOP: intraocular pressure, CMT: central macular thickness, DR: diabetic retinopathy, NPDR: non-proliferative diabetic retinopathy, PDR: proliferative diabetic retinopathy, VEGF: vascular endothelial growth factor, DRIL: disorganization of the retinal inner layers.

**Table 2 medicina-59-01125-t002:** Univariate analysis of recurrence interval.

	≥12 Weeks (*n* = 8)	<12 Weeks (*n* = 10)	*p* Value
Age (years) (mean ± SD)	72.3 ± 8.3	66.0 ± 10.4	0.1837
Sex, male/female	2/6	5/5	0.2796
Duration of diabetes (years) (mean ± SD)	16.9 ± 12.3	18.0 ± 15.2	0.8655
HbA1c (%) (mean ± SD)	7.2 ± 0.6	8.0 ± 1.3	0.1520
Concomitant hypertension, +/–	5/3	4/6	0.3406
Concomitant hyperlipidemia, +/–/unknown	3/1/4	2/4/4	0.1889
Concomitant of chronic kidney disease, +/–/unknown	2/4/2	2/1/7	0.3406
Lens status, phakia/pseudophakia	1/7	3/7	0.3641
BCVA at baseline (logMAR) (mean ± SD)	0.14 ± 0.26	0.30 ± 0.29	0.2457
IOP at baseline (mmHg) (mean ± SD)	15.8 ± 3.1	16.5 ± 2.8	0.5746
CMT at baseline (µm) (mean ± SD)	377.6 ± 215.5	550.8 ± 205.3	0.1097
DR status, mild NPDR/moderate NPDR/severe NPDR/PDR	2/0/6/0	2/1/3/4	0.0520
Duration of DME (months) (mean ± SD)	14.8 ± 5.2	27.8 ± 8.8	0.2309
Number of previous anti-VEGF injections (mean ± SD)	3.9 ± 3.3	7.1 ± 6.2	0.2106
Type of previous anti-VEGF injections, ranibizumab/aflibercept/both	1/5/2	2/4/4	0.6351
Complications with vitreous traction and/or epiretinal membranes	3 (37.5%)	6 (60.0%)	0.3406
DRIL	1 (12.5%)	6 (60.0%)	0.0326 *
Disorganization of ellipsoid zone	4 (50.0%)	6 (60.0%)	0.6713
Prior treatments other than anti-VEGF therapy			
Subtenon injections of corticosteroids	2 (25.0%)	9 (90.0%)	0.0034 *
Focal macular laser	6 (75.0%)	4 (40.0%)	0.1316
Pars plana vitrectomy	0	3 (30.0%)	0.0897

*: *p* < 0.05, DME: diabetic macular edema, SD: standard deviation, HbA1c: glycated hemoglobin, BCVA: best-corrected visual acuity, logMAR: logarithm of the minimum angle of resolution, IOP: intraocular pressure, CMT: central macular thickness, DR: diabetic retinopathy, NPDR: non-proliferative diabetic retinopathy, PDR: proliferative diabetic retinopathy, VEGF: vascular endothelial growth factor, DRIL: disorganization of the retinal inner layers.

## Data Availability

Data sharing is not applicable to this article.

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
