# Peer review of "Faricimab for Diabetic Macular Edema in Patients Refractory to Ranibizumab or Aflibercept"

_medicina, 2023, doi:10.3390/medicina59061125_

Round 1
Reviewer 1 Report
Topic of benefit of switch to faricimab from other anti-VEGF agents of significant interest as the field learns about the benefits of this new agent. This manuscript attempts to provide new details about the benefits of switching to faricimab, but the study's very small sample size limits the strength of the conclusions. The authors found the treatment interval could be lengthened and the vision was no different with faricimab. A major flaw of the study design is the poor characterization of the decision to treat. Treatment was prn, and the decision to treat was based on new fluid. The authors should quantify, if possible, how much change prompted treatment. The currently described protocol sounds fraught with bias, as the treating physician may be quicker to overlook some new cysts of fluid with faricimab as he/she was with ranibizumab. The prn protocol should also be better characterized - how often were patients seen, etc.
English language editing recommended. Example grammatical errors:
because of the inhibit of Ang-2 – page 3
the combined dexamethasone/ranibizumab group was shown to cause significantly larger functional and anatomical improvements – page 9
rather than to vascular permeability – page 9
presence and greater severity of DRIL were poor predictors of the BCVA – page 9 , do you mean predictors of poor BCVA? The articles you cite state DRIL is a significant predictor of vision outcomes
Reviewer 2 Report
The authors present results of using Faricimab for diabetic macular edema. This novel antiVEGF has been recently introduced in clinical practice. In fact, it has not been approved in all countries yet, although it will in short time.
The authors explain in detail the pathophysiology of this antiVEGF. Introduction is adequate, methods are presented clearly, results are well presented, and the discussion section is deep enough. A deep revision has been performed, and conclusions are fully supported by the presented results. Nevetheless, the main issue is the number of patients included in the analysis; 18 patients are too few. Outcomes of clinical trials include higher sample sizes. Therefore, even though the manuscript is very interesting and novel, I would like to encourage the authors to include some more patients. They included patients between July and October 2022; maybe they could add some more patients treated between November and April. I am sure that those outcomes would be really interesting for all retina specialists worldwide.
Round 2
Reviewer 1 Report
Acceptable edits
Acceptable edits
Reviewer 2 Report
I acknowledge the authors for their efforts searching for more patients treated with intravitreal Faricimab. It is a pity that no more patients received this treatment despite being a major hospital in Japan. Additionally, no comparisons have been made with any other intravitreal drugs, so outcomes have little use in daily practice.